# Microbially-Induced Calcium Carbonate Precipitation Test on Yellow Sandstone Based on LF-NMR Monitoring

**DOI:** 10.3390/ijerph192416860

**Published:** 2022-12-15

**Authors:** Chao Zhuang, Chuang Liu, Ziteng Cui, Ze Yang, Yongqiang Chen, Zhi Dou

**Affiliations:** 1School of Earth Sciences and Engineering, Hohai University, Nanjing 210098, China; 2Qingdao Zhongyu Environmental Testing Co., Ltd., Qingdao 266600, China

**Keywords:** microbial induction, calcium carbonate, nuclear magnetic resonance, yellow sandstone, pore structure

## Abstract

The microbially-induced calcium carbonate precipitation (MICP) technique has shown great robustness in dealing with soil and groundwater contamination problems. A typical result of the implementation of MICP technique is a change in the pore structure. In this study, the effects of MICP on the pore structure of yellow sandstone from the Zigong area, Sichuan, China under different conditions, (e.g., temperature, pH, and calcium ion concentration) are investigated using LF-NMR resonance. The pore network of yellow sandstone is accurately measured using the peak area of the *T*_2_ spectral signal. The distribution of calcium carbonate in the pores of the yellow sandstone is characterized by the magnitude of the *T*_2_ signal variation. The results show that the precipitation of calcium carbonate caused by MICP tends to be deposited in relatively large pores. However, the calcium carbonate precipitates in the smaller pores at a higher temperature. A higher pH considerably enhances the precipitation, and the alkaline environment tends to cause the precipitation of the calcium carbonate in the large pores. Although the amount of produced calcium carbonate continuously increases as the MCIP process continues, which is expected, the production efficiency decreases steadily.

## 1. Introduction

In recent years, there has been a surge of interest in the soil modification technology known as microbiologically-induced calcium carbonate precipitation (MICP) [1]. To prevent the liquefaction of sandy soils, soil erosion, and natural erosion, the MICP technique primarily fills the fractures, cracks, and pores in materials with cross-linkable mineral components (e.g., calcium carbonate) produced by bacterial metabolism [2,3]. Compared to conventional physicochemical modification techniques, MICP is easier to implement, less costly to operate, produces almost no toxic or hazardous substances, and has less chemical impact on the soil and water environment [4]. Meanwhile, compared to conventional chemical slurries, the bacterial and cementing fluids used in this technology are less viscous, can handle greater thicknesses and deeper levels of rock mass, and are easier to incorporate into the geotechnical materials [5,6,7,8].

The pore structure of sandstone, a typical porous medium, is critical for the diffusion, seepage, and transport of fluids. The pore distribution in sandstones is typically heterogeneous, discontinuous, and anisotropic, making the accurate characterization of its pore structure very difficult [9,10]. Low-field nuclear magnetic resonance (LF-NMR) as a non-destructive, rapid, and accurate technique for characterizing the pore structure of rocks has become increasingly popular [11,12,13]. Measuring the relaxation time of hydrogen-containing fluids (methane, water, and diesel) in rocks under the magnetic field allows this technique to accurately characterize the pore structure and pore fluid distribution [14,15,16,17].

The physical characterization of rock masses has been gradually integrated into the application of LF-NMR after being initially used for petroleum logging. Information on the rock porosity, pore size distribution, pore connectivity, free/bound fluid saturation, and permeability can be obtained from the LF-NMR relaxation *T*_2_ spectra [12,18,19,20,21,22].

The impacts of the calcium ion concentration, temperature, pH, calcium ion concentration, and soil particle size on the curing effect in the MICP reaction are the key influences on MICP. Sun et al. [23] pointed out that temperature had a more significant impact on enzyme activity than it did on the rate of growth of the typical bacteria employed in MICP. Amo et al. [24] pointed out that the pH value has a great influence on the morphology of calcium carbonate particles synthesized at room temperature and a high calcium ion concentration tends to limit microbial activity [25]. According to Owthaman et al. [26] and Lin et al. [27], the MICP effect was impacted by the varying contact areas and solution flow rates that came from the different particle sizes of the various soils. The above studies have shown that MICP can significantly increase the soil strength, while it can decrease the permeability. This is due to the fact that the solid precipitation created by the reaction can tie the loose soil fragments and fill the solid pore structure, decreasing the soil porosity and the infiltration rate. The majority of the current studies are concentrated on the impact of the external factors on the uniformity of the healing effect, while the fluctuation of pore space within the rock mass has not attracted attention. Detailed studies are still needed to determine the feasibility and modification process of the microscopic characterization of pore structure using MICP technology [28,29]. The following research was conducted to describe how MICP can be used to change the pore structure of yellow sandstone.
The LF-NMR approach was used to test the changes in the pore structure of yellow sandstone before and after the use of MICP technique.The filling effect of MICP on various pores under experimental temperature, pH, and calcium ion circumstances was studied.We compared the MICP process’s calcium carbonate yield and production efficiency under various testing settings.

## 2. Materials and Methods

### 2.1. Experimental Materials

#### 2.1.1. Experimental Samples

The specimens were chosen from yellow Upper Jurassic sandstone in the Zigong district of Sichuan, China, which is rich in oxides and contains only a small quantity of iron and which is predominantly granular in structure and reasonably compact. It has a density of 2.589 g/cm^3^ at room temperature. The specific samples were cylindrical samples with a diameter of 1 inch and a length of 2 inches. The rocks were cleaned of impurities by soaking them in 0.1 M hydrochloric acid solution for 12 h, and then, they were dried.

#### 2.1.2. Preparation of the Bacterial Suspension

The bacterium Sporosarcina pasteurii, a gram aerobic bacterium (ATCC11859), was used in this study. By adding the frozen bacteria to the Luria-Bertani medium (LB) and shaking the incubator at 30 °C for 48 h, the frozen bacteria were activated. The activation was successful when the bacterial suspension was noticeably turbid [30].

#### 2.1.3. Solution of the Calcium Source

The liquid environment for MICP was a calcium source solution (55.5 g/L CaCl_2_, 10 g/L NH_4_Cl, and 30 g/L urea). Bacillus can develop a urease that promotes the organism’s natural metabolic process for external urea hydrolysis. Since the bacteria are negatively charged, the calcium ions in the environmental solution are adsorbed on the cell surface. 

#### 2.1.4. Experimental Setup

The MICP experimental setup is shown in Figure 1.

The sandstone was treated with hydrochloric acid during the washing and drying before being immersed in a bacterial suspension (OD_600_ = 0.6) for two hours. It was then transferred to a beaker containing a calcium solution and subjected to a MICP mineralization test in a constant temperature oven for 48 h. The rocks were finally rinsed with deionized water and dried in an oven at 110 °C for 12 h. An oxygen pump was added to the calcium solution to supply more oxygen, and the bottom was placed in a magnetic stirrer to evenly distribute the liquids in the concentration of the system.

The experiments were conducted using a technique in which the entire yellow sandstone was infiltrated into a bacterial suspension due to the rapid reaction time. The advantage of this approach is that there is no need to replenish fresh calcium solution and drain the original waste solution as the calcium solution can freely infiltrate into the interstitial spaces of the sample [31]. The disadvantage of this method is that the produced calcium carbonate could be deposited on the surface layer of the sample and clog the circulation pores, making the mass transfer of nutrients and metabolic waste between the sample and the environmental solution impossible. This reduces the effectiveness of the MICP [32].

### 2.2. Test Method

#### 2.2.1. Concentration and Activity of the Bacterial Suspension

The absorbance of the liquid at 600 nm is expressed as OD_600_. Usually, this technique is used in laboratories to measure the concentration of bacterial suspensions. According to this theory, the concentration of light-absorbing substances in the solution determines the absorbance value, and the concentration of bacterial cells determines the turbidity of the bacterial solution. After activation, the higher concentration of bacterial suspension is diluted to an appropriate OD_600_ value by LB, and it is used as the experimental bacterial suspension [33].

The method of conductivity of the solution change per unit time was proposed to characterize the bacterial activity. According to the mechanism of urea hydrolysis reaction, the ammonium and carbonate ions generated by the reaction increase the conductivity of a solution. In one experiment, the bacterial suspension and urea solution were combined and the change in conductivity of the solution was monitored over a three minute period using a meter to measure the conductivity of the solution to calculate change in the average conductivity of the solution per minute (mS/min). Eleven mM urea hydrolyzed per minute corresponds to a conductivity of the solution change of 1 mS per minute. Therefore, the amount of urea hydrolyzed by the urease per unit time can be calculated using the average conductivity of the solution change per minute (mS/min).

#### 2.2.2. Experiments on the MICP Mineralization

Three parameters—temperature, pH, and calcium ion concentration—were established for the experimental MICP tests to investigate the changes in the pore structure of calcium carbonate caused by MICP exposure in yellow sandstone under different conditions. Table 1 shows the experimental conditions for each group.

#### 2.2.3. Nuclear Magnetic Resonance Testing

The equipment used for this test is the MacroMR12-150H-I large-bore LF-NMR analyzer which was developed by Suzhou Newmark. The diameter of the probe coil was 25 mm, the radio frequency power was 300 W, the temperature of the permanent magnet was 32.0 °C, and the resonance frequency was 13.05 MHz. The experimental equipment is shown in Figure 2.

The test system stimulates Seeman splitting in the ^1^H nucleus using the physics of magnetic resonance of the hydrogen atoms that are excited by RF pulses of a specific frequency in a magnetic field [34,35]. The moisture content and distribution state of the sample were determined by analyzing the *T*_2_ spectra, and the proton density and distribution within the sample was determined by examining the water ^1^H [36,37].

In the experiments, the yellow sandstone was placed under a vacuum before and after the MICP test, which was saturated at a pressure of 10 MPa for 12 h, and then, it was placed in the LF-NMR coil to test the precipitation of calcium carbonate. Therefore, the saturated samples were wrapped in cling film before testing to prevent the loss of saturated water. The influence of the signal from the cling film itself on the LF-NMR test results was not considered because it is weaker than the signal from the water in the rock in the LF-NMR measurement [38,39].

#### 2.2.4. Calcium Ion Content Test

Calcium ion utilization may be affected by the residual calcium ion concentration in the lateral solution. Therefore, the experiments were conducted to find out how much calcium carbonate was deposited in the rock by performing a calcium ion titration test on the solution that remained after mineralization. The ethylene diamine tetraacetie acid (EDTA) titration of the calcium ion concentration was used as the test method.

## 3. Results and Analysis

### 3.1. Effects of Temperature

According to the reaction mechanism of urea hydrolysis, as they are hydrolysis products, the formation of ammonium and carbonate ions increases the conductivity of the surrounding solution. In one experiment, 18 mL of 1.6 M urea solution with OD_600_ values of 0.6, 0.8, and 1.0 was injected into 2 mL of a bacterial suspension at 25, 30, 35, 40, and 45 °C. The urease activity was determined from the average number of conductivity changes per minute.

Urease activity is the most important temperature factor affecting the MICP process. Since urease is basically a protein, the temperature at which it functions best is called the optimal urease temperature. When the temperature falls below the optimal value, the activity of urease decreases until it reaches a point where its catalytic efficiency is zero and it is inhibited. The activity of urease gradually increases to its peak when the temperature reaches its optimum value. As the temperature increases, the activity of urease decreases rapidly above the optimal temperature, and at a certain point, it is inactivated by denaturation. Even if the ideal temperature was regained at this point, the urease would not be able to do so. According to Figure 3, the bacterial activity increased steadily with an increasing temperature, with a clear tendency to grow at 25–45 °C.

Within the theoretical concept of the *T*_2_ spectra, the *T*_2_ spectral curves of the samples before and after MICP have only three peaks at different temperatures, indicating that the range of the pore size distribution of the yellow sandstone samples does not change after the MICP, and the changes are mainly in the distribution ratio of different pore sizes. According to Figure 4, the four signal curves after the experiment showed no difference between the first peak of the original sample signal and the four signal curves after the experiment. The breakthrough occurred at the second peak, and the signal at the third peak decreased significantly with increasing temperature, indicating that the large pores were filled by the precipitation of calcium carbonate after the MICP, resulting in the medium pores, while MICP had less influence on the small pores.

### 3.2. Effect of pH

The electrical charge within the organism, the cell membrane charge, and the uptake and utilization of nutrients by microorganisms are all affected by the changes in the pH, which in turn affects bacterial activity. Two mL of bacterial suspensions with OD_600_ values of 0.6, 0.8, and 1.0 were injected into eighteen mL of 1.6 M urea solution at pH = 6, 7.5, and 9, respectively, and the change in conductivity per minute was determined to investigate the effect of the pH on the bacterial activity. Figure 5 shows the bacterial activity under different pH conditions.

According to Figure 6, due to the different production batches, the basal signals of the yellow sandstone samples vary slightly. On the other hand, the absence of a noticeable break in the middle of each peak indicates that the pores are essentially continuous. When the pH of the solution was increased, the effect of filling the large pores of the yellow sandstone sample was very clear compared to the experimental results at different temperatures. In other words, compared to the area of the first peak, the area of the second peak representing the large pore size changed drastically.

### 3.3. Effect of Calcium Ion Concentration

The starting material for the mineralization that leads to calcium carbonate precipitation is the calcium ions in the calcium source solution [40]. This method of determining bacterial activity by testing the average value of conductivity of the solution changes is not possible because the calcium ions in the solution combine with the carbonate ions produced by urea hydrolysis to form calcium carbonate precipitates that affect the conductivity of the solution. Therefore, the tests were conducted to determine the bacterial activity using the technique of determining the NH^4+^ concentration of the urea hydrolysis products [41]. The experimental setups with 0.5, 1.0, 1.5, and 2.0 M calcium chloride and urea mixtures with 1 M urea were used to test the NH^4+^ concentrations before and after a 60 min reaction.

Figure 7 shows how the bacterial activity decreases significantly with increasing amounts of calcium ions in the solution, illustrating the clear inhibitory effect of high salt concentrations on the bacterial activity. However, the production of calcium carbonate is increased by high concentrations of calcium ions. This indicates that increasing the concentration of calcium ions in the solution is a viable method to increase the precipitation of calcium carbonate, but it should be controlled within a certain concentration, otherwise the rate of hydrolysis will decrease, leading to a decrease in the experimental efficiency.

Figure 8 shows the distribution of the *T*_2_ spectra of the yellow sandstone produced by the MICP in an environment with different calcium ion concentrations. The third peaks of the other three groups almost disappeared at a calcium ion concentration of 0.5 M, indicating that the number of large pores was significantly reduced. At a calcium ion concentration above 1 M, some of the large pores disappeared, indicating that the range of the pore size distribution changed. The precipitation effect of calcium carbonate in the presence of a high concentration of calcium ions is the main cause of this phenomenon. The small pores hardly changed, while the number of medium pores decreased to varying degrees. The total peak area decreased significantly overall, indicating that there are significantly fewer pores overall after the MICP mineralization. The larger the pores are, the easier it is to fill them.

### 3.4. Pore Size Variation Trend

The pores with transverse relaxation times of less than 10 ms in the *T*_2_ spectra are classified as small pores, those between 10 and 100 ms are classified as medium pores, and those with more than 100 ms are classified as large pores, which was conducted in order to analyze the distribution of the pore structure within the rock sample after the experiment more precisely.

The distribution of the pores in the yellow sandstone are shown in Table 2 both before and after MICP. The overall results show that the degree of change caused by MICP increases with the pore radius. As the temperature increases, the water content of the sample gradually decreases, the area of the first peak gradually shrinks, and calcium carbonate tends to precipitate in the small pores. The area of the *T*_2_ spectrum corresponding to the large pores shrank significantly at a higher solution pH, while the amount of signal corresponding to the small and medium pores remained essentially unchanged. This is due to the fact that the carbonate ions generated by the hydrolysis of urea rapidly combined with the calcium ions in the large pores. This indicates that calcium carbonate tends to fill the pore volume with an increasing calcium ion concentration. The total *T*_2_ spectrum of the sample gradually decreased as the concentration of calcium ions in the solution increased. The filling effect of the sample was very clear, with there being less precipitation in the small pores, more precipitation in the medium pores, and highest amount of precipitation in the large pores.

Figure 9 compares the porosity of the sandstone samples before and after the MICP experiments under different conditions. For the samples used for the experiment with 0.5 M calcium ions, the change in porosity before and after the experiment was smaller than it was for the other samples. The main reason for this is that calcium carbonate production has a smaller effect on porosity when it is low. In addition, the volume of the biofilm itself becomes the main factor affecting the porosity since the biofilm produced during bacterial growth and multiplication occupies the pore volume of the samples [8,42].

### 3.5. Calcium Carbonate Production and Utilization

The amount of calcium ions in the solution can be used to identify the calcium carbonate formed in the sample by monitoring the MICP reaction, in which the calcium ions in the solution continuously combine with the carbonate ions formed by the hydrolysis of urea to form calcium carbonate precipitates in the pore structure. The efficiency of calcium ion utilization and the formation of calcium carbonate are shown in Figure 10. The yellow sandstone used in the experiment could not be easily destroyed. Instead, the amount of calcium carbonate formed in the sample was estimated by measuring the remaining solution volume and calcium ion concentration after the MICP exposure.

The results showed that the temperature, pH, and calcium ion concentration have a significant effect on the production of calcium carbonate. The nutrient composition of the solution and bacterial activity are the main factors explaining this phenomenon. The results of the experiments at different temperatures indicate that the influence of temperature on the bacterial activity leads to a faster exchange of substances inside and outside the cell, resulting in a higher utilization of calcium ions and nutrients under the same conditions. However, in conjunction with the information on bacterial activity at different temperatures, it was found that an increase in the temperature did not lead to a steady increase in the calcium carbonate production at the same calcium ion concentration. Additionally, in previous studies, the rate of the hydrolysis of urea was found to have a significant effect on the type of calcium carbonate crystals produced by MICP. Faster hydrolysis rates typically produce amorphous calcium carbonate, while slower hydrolysis rates tend to produce relatively stable crystals such as calcite [43]. Spherical aragonite will form at supersaturation, and because of its sub-stability and high solubility area, it will take up less space in the voids and have an impact on the experimental findings. As a result, the test was performed after waiting for it to change into a stable calcite with a lower solubility area. Therefore, the microbially-induced precipitation of calcium carbonate at higher temperatures is not more advantageous. By combining the results of the analyses of calcium carbonate production, it was found that a temperature of approximately 30 °C was more suitable for MICP actions to occur. This is generally in accordance with Peng’s findings, which show that temperature can influence the MICP response by affecting the rate of bacterial growth, urease activity, and calcium carbonate synthesis [44,45]. There was a little correlation between the effects of calcium carbonate production under the influence of different pH values and the results of bacterial activity tests. This could be due to the fact that electrochemical reactions prevail in the alkaline environment where the calcium carbonate is deposited. As a result, the calcium and carbonate ions produced by urea hydrolysis bind together rapidly, increasing the effectiveness of calcium ion utilization. The higher the concentration is, the more material that can be used for precipitation, and the easier it is to form calcium carbonate precipitates, as shown by the MICP experiments at different calcium ion concentrations. However, the high salt concentration environment inhibits the rate of urea hydrolysis or the activity of bacteria, and not all of the calcium ions in the solution can be utilized. Therefore, the utilization of calcium decreases with an increasing calcium ion concentration.

Contrary to the empirical expectation that bacteria tend to adsorb in small pores, the experimental results show precipitation in the large pores. The author considers several possible factors, including that the small pores of the yellow sandstone used in the experiment prevented the bacteria from being fully distributed in the sample by osmosis, and while the particular nature of the bacteria prevented them from being saturated by high pressure devices, some of the small pores are inaccessible to the bacteria. Secondly, the surface area of the large pores is larger than that of the small pores, so they can hold more enriched bacteria. In addition, the abundance of nutrients in the large pores facilitates the precipitation of calcium carbonate. Additionally, as the test procedure necessitates the excitation of moisture in the rock void, extending the test period will raise the sample’s temperature, which will result in a test error. Since the NMR signal describes the sample’s water content, any remaining moisture on the sample’s surface will also result in some test mistakes. In conclusion, MICP-induced alterations in the pore space of yellow sandstones, whose pore structure changes are closely connected to the experimental circumstances, can be successfully characterized by NMR testing. At the same time, there is a large difference in the effect of different experimental conditions on the MICP effect, and this difference is also reflected in the pore structure variation.

## 4. Conclusions

In this study, the temperature, pH, and calcium ion concentration of the experiment were changed to perform MICP experiments on yellow sandstone. Before and after the experiments, the bacterial activity of the samples, the calcium ion concentration and LF-NMR were tested, and the effects of MICP action on the pore structure of yellow sandstone from the Zigong area under different conditions were investigated. The results led to the following conclusions:

(1) The proportional distribution of different pores in the sample can be seen in the NMR *T*_2_ spectra, and the smaller the chirality time is, the smaller the pore radius is. The trend in the distribution of calcium carbonate in the pores created by MICP can be accurately depicted by comparing the *T*_2_ spectral distribution of studies conducted on yellow sandstone. After the MICP experiment, the number of large pores decreased significantly, and the proportion of medium and small pores increased.

(2) The distribution trend of calcium carbonate precipitation in yellow sandstone is affected by the experimental conditions. The increase in the temperature leads to an increase in the calcium carbonate precipitation in the small pores, the alkaline environment concentrates calcium carbonate production in the large pores, the concentration of calcium ions has a significant effect on the amount of calcium carbonate precipitation in the large and medium pores, and it changes the range of pore size distribution in large pores, and the number of occupied pores increases significantly with an increasing concentration. 

(3) The amount of calcium carbonate produced is directly governed by the bacterial activity, and this in turn affects the porosity of the sample. This correspondence varies with the MICP effect results under different experimental arrangements. The calcium carbonate precipitation at a higher pH is dominated by electrochemistry, and the effect of bacterial activity on calcium carbonate production is relatively weak due to the less acidic and alkaline environment during the bacterial activity, the increase in calcium carbonate precipitation at lower temperatures affects calcium carbonate production, and the decrease in the bacterial activity at higher temperatures decrease calcium carbonate precipitation.

## Figures and Tables

**Figure 1 ijerph-19-16860-f001:**
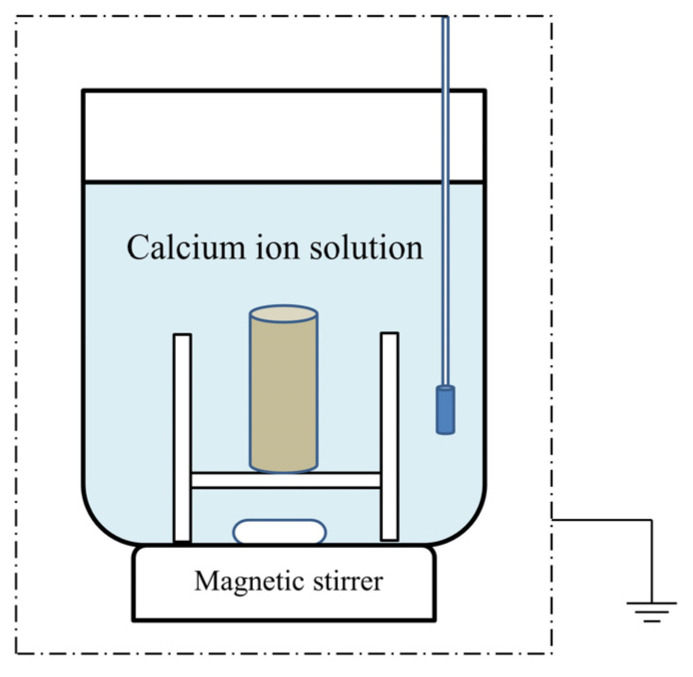
Schematic diagram of the MICP experimental setup.

**Figure 2 ijerph-19-16860-f002:**
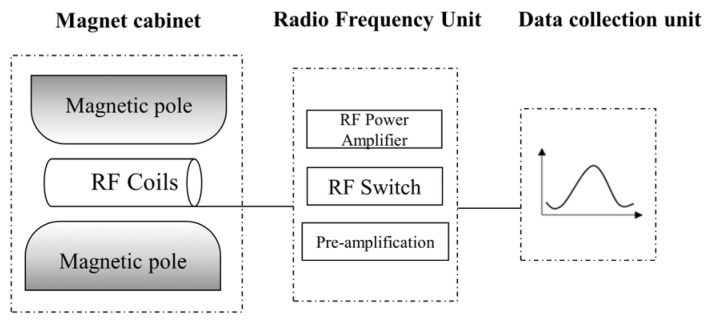
Schematic diagram of the nuclear magnetic resonance analyzer.

**Figure 3 ijerph-19-16860-f003:**
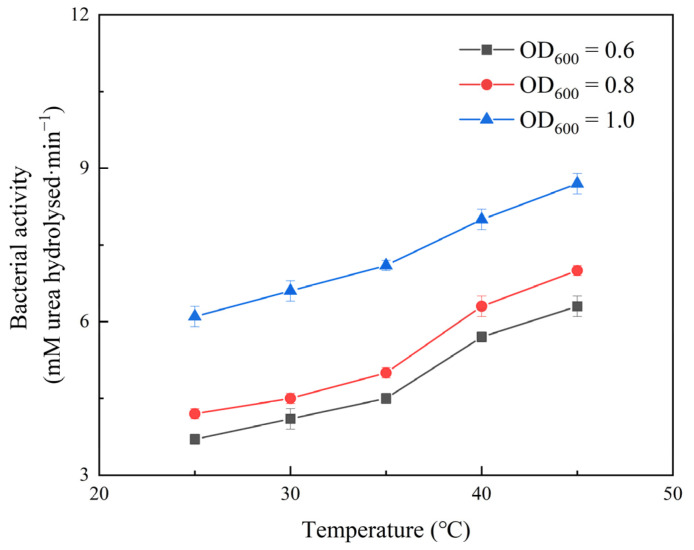
The effect of temperature on bacterial activity.

**Figure 4 ijerph-19-16860-f004:**
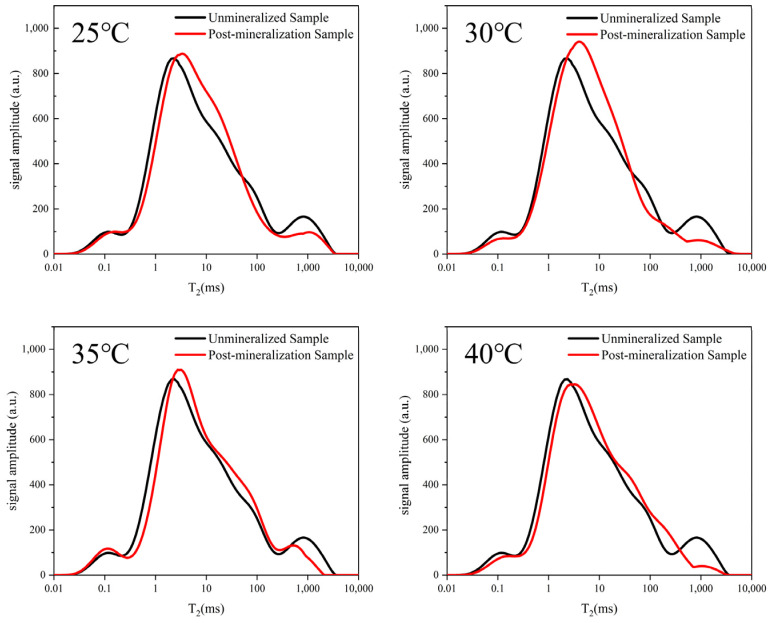
Temperature effects on the *T*_2_ spectra of MICP-interacting yellow sandstone.

**Figure 5 ijerph-19-16860-f005:**
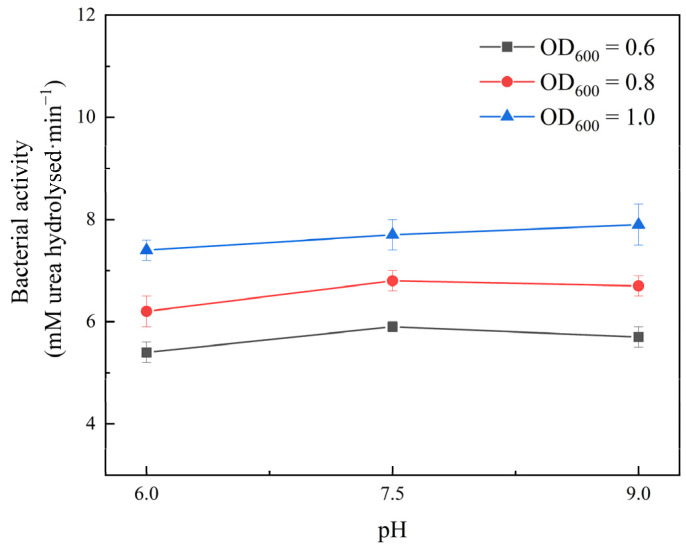
Effect of pH on bacterial activity.

**Figure 6 ijerph-19-16860-f006:**
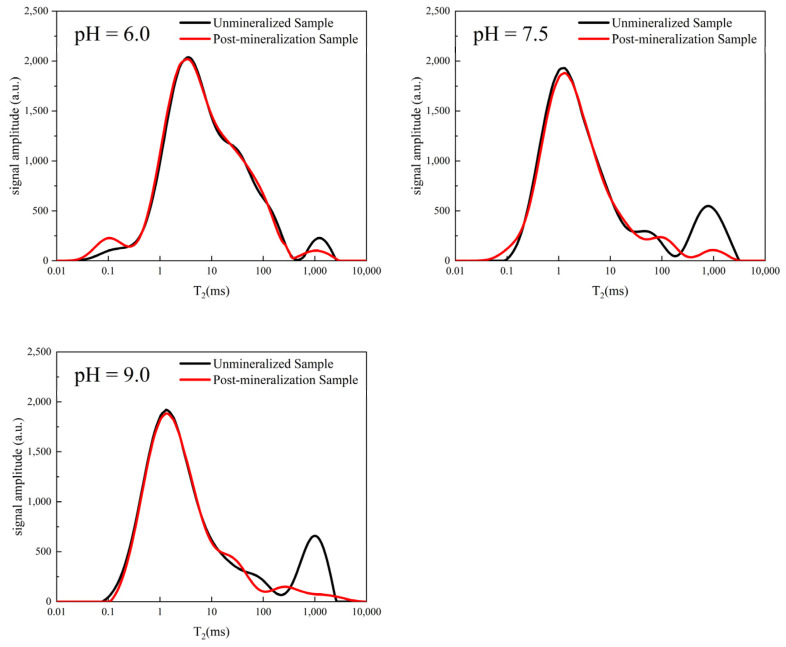
pH effects on the *T*_2_ spectra of MICP-interacting yellow sandstone.

**Figure 7 ijerph-19-16860-f007:**
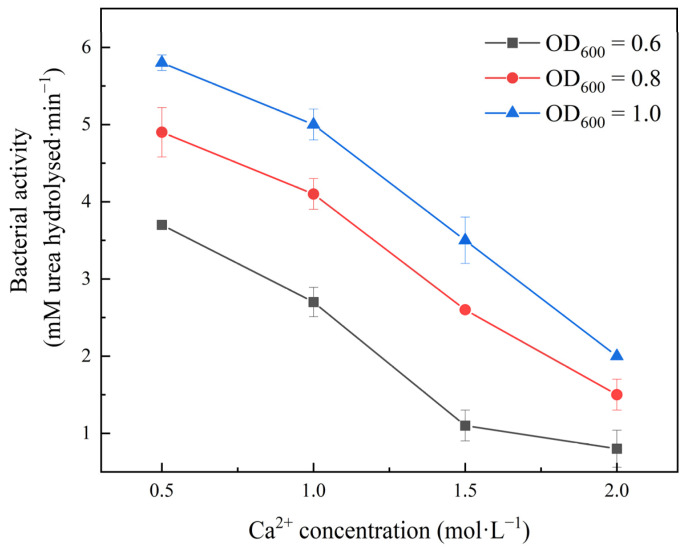
The effect of calcium ion concentration on bacterial activity.

**Figure 8 ijerph-19-16860-f008:**
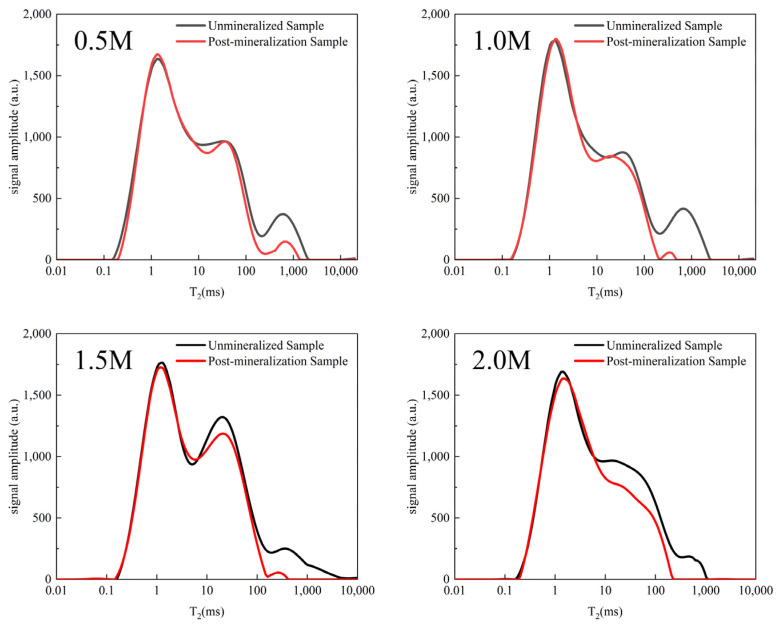
Calcium ion concentration effects on the *T*_2_ spectra of MICP-interacting yellow sandstone.

**Figure 9 ijerph-19-16860-f009:**
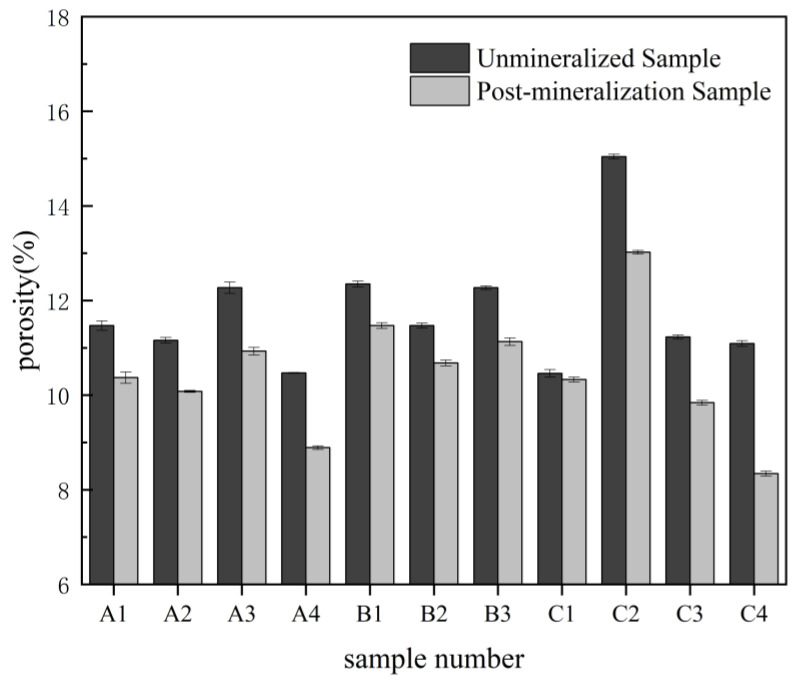
The change of porosity.

**Figure 10 ijerph-19-16860-f010:**
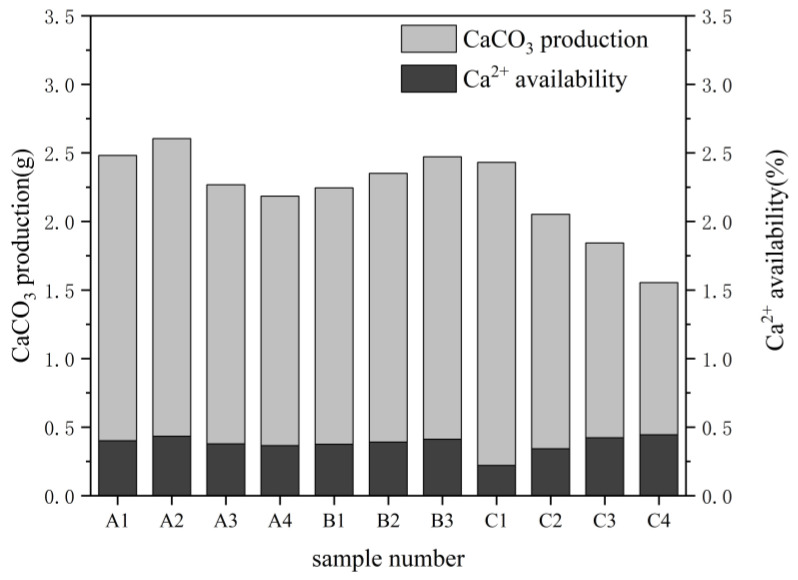
Calcium carbonate production and calcium ion utilization.

**Table 1 ijerph-19-16860-t001:** Experimental parameters.

Experimental Group	Variable Settings	Other Experimental Conditions
Temperature	25 °C	pH = 7.5 Ca^2+^ 1.0 M
	30 °C	
	35 °C	
	40 °C	
pH	pH = 6.0	30 °C Ca^2+^ 1.0 M
	pH = 7.5	
	pH = 9.0	
Ca^2+^ concentration	0.5 M	30 °C pH = 7.5
	1.0 M	
	1.5 M	
	2.0 M	

**Table 2 ijerph-19-16860-t002:** Percentage of pore diameter before and after mineralization.

Experimental Group	Sample Number	Before Mineralization			After Mineralization		
		Small Pores %	Middle Pores %	Large Pores %	Small Pores %	Middle Pores %	Large Pores %
Temperature	A1	12.58	72.06	15.36	16.79	70.9	12.31
	A2	13.55	71.69	14.76	18.34	70.74	10.92
	A3	12.99	71.20	15.81	17.84	70.92	11.24
	A4	14.11	70.97	14.92	18.34	73.45	8.21
pH	B1	14.97	74.58	10.45	16.21	75.12	8.67
	B2	10.68	74.96	14.36	12.75	79.58	7.67
	B3	11.01	73.2	15.79	14.21	80.47	5.32
Ca^2+^ concentration	C1	9.06	74.73	16.21	11.06	78.78	10.16
	C2	8.87	73.04	18.09	10.21	81.81	7.98
	C3	8.46	76.88	14.66	9.87	79.1	11.03
	C4	9.21	77.3	13.49	10.26	79.87	9.87

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
