# Peer review of "Microbially-Induced Calcium Carbonate Precipitation Test on Yellow Sandstone Based on LF-NMR Monitoring"

_ijerph, 2022, doi:10.3390/ijerph192416860_

Round 1

Reviewer 1 Report

Review report on ijerph-2066138
Microbially induced calcium carbonate precipitation test on yellow sandstone based on LF-NMR monitoring
Chao Zhuang, Chuang Liu, Zhi Dou *, Ziteng Cui, Ze Yang, Yongqiang Chen

The manuscript studied the implementation of microbially induced calcium carbonate precipitation (MICP) in pore structure change in yellow sandstones. The pore structure under different geochemical conditions were characterized by the peak area of the T2 spectral signal via LF-NMR resonance. This paper is interesting for applying MICP to test the pore structure change. From my point of view, this manuscript has the potential for publication in ijerph, but needs some revisions especially in introduction section.
I suggest some points needed to be improved. A number of suggestions are given, but not limited to the following:

1.     The sequence of reference needs to be checked carefully. In introduction, line 26, the number of references started from 6, then went to 23 in line 29, which is not correct.

2.     Line 48: ‘The above studies have shown that MICP can significantly increase soil strength and permeability’, because of …

3.     In introduction, the authors failed to provide a comprehensive literature review in this relevant area. I believe there could be tons of work testing the MICP controlled pore structure changes at different environmental conditions. It is necessary to give a summarization on previous works.

4.     Following the aforementioned comment, the authors need to clearly highlight the novelty/importance of this work. What is different/unique of this MS compared to existing works? How can your work contribute to the existing knowledge?

5.     In section 2.2.1, it is important to show the sample mineralogy. Although quartz is expected to be the main composition of specimen, it may still contain certain fractions of carbonate minerals as cement or pore-filling area, which may affect results and analyses.

6.     Overall, the results are straightforward to follow and understand. It is recommended to discuss the measurement error due to limitation of the instrument.

Author Response

We thank the reviewers for taking the time to review our manuscripts. We have responded to each point in the annex. For clarity, comments are in black and our responses are in red.

Reviewer 2 Report

General comments

Research that fits the scope of the journal, and has been presented with a detailed methodology. Despite this, the authors need to provide more details before publication

Specific comments

Lines 11-12. I suggest to change “structure” into “network”

Lines 25-55. Provide more details on the yellow sandstone on: “the geological nature”, “petrophysical properties”, and the conceptual link with the soil processes that you mention

Lines 25-55. Explain why you decided to test such a sandstone

Lines 36-39. “The pore structure of sandstone, a typical porous medium, is critical for diffusion, seepage, and transport of fluids. The pore distribution in sandstones is typically heterogeneous, discontinuous, and anisotropic, making accurate characterization of pore structure very difficult”. Add recent and relevant review paper on anisotropies and heterogeneities on sandstones:

- Medici, G. and West, L.J., 2022. Review of groundwater flow and contaminant transport modelling approaches for the Sherwood Sandstone aquifer, UK; insights from analogous successions worldwide. Quarterly Journal of Engineering Geology and Hydrogeology. 55, 4, https://doi.org/10.1144/qjegh2021-176

Line 11. Add age and location of the Yellow Sandstone. Here and in the abstract

Line 47. Add relevant and specific literature on NMR/T2 signal in sandstone, rivers of papers come from the hydrocarbon industry sector, see example below:

- Rosenbrand, E., Fabricius, I.L., Fisher, Q. and Grattoni, C., 2015. Permeability in Rotliegend gas sandstones to gas and brine as predicted from NMR, mercury injection and image analysis. Marine and Petroleum Geology64, pp.189-202.

- Hossain, Z., Grattoni, C.A., Solymar, M. and Fabricius, I.L., 2011. Petrophysical properties of greensand as predicted from NMR measurements. Petroleum Geoscience17(2), pp.111-125.

Line 55. Add also the three specific objectives of your research using numbers (e.g., i, ii, and iii)

Line 59. What it is the Yellow Sandstone? A specific geological formation (if yes use capital letters), or just a sandstone of yellow colour?

Line 72. All the equations are necessary? All have been recalled later in the text?

Line 101. Please, be more specific when you say “conductivity”. Fluid or rock conductivity? Which type of conductivity (e.g., electric, hydraulic etc. etc.)?

Line 192. “it is not possible”. Please, do not start a new sentence with “it”. Revise the structure of the sentence

Line 273. Possible to compare your results also to other researches?

Line 288 “Consider the following factors”. Sentence too short, and disconnected with the previous and the next one. Please, revise the language and avoid repetitions of words

Lines 302-322. You present three conclusive points. Therefore, you should state the three specific objectives of your research in your introduction. See my comment above

Line 323. Add a closing sentence at the end the summarizes the “take home message” of the paper

Line 330. Add the relevant and recent references suggested above

Figures and tables. Increase graphic resolution of Figures 1 and 2 by rising the number of dpi. The quality of these two figures is lower than the others

Author Response

We thank the reviewer for taking the time to review our manuscripts. We have responded to each point in the annex. For clarity, comments are in black and our responses are in red.
